# All-Fiber CO_2_ Sensor Using Hollow Core PCF Operating in the 2 µm Region

**DOI:** 10.3390/s18124393

**Published:** 2018-12-12

**Authors:** Sully Milena Mejia Quintero, Luiz Carlos Guedes Valente, Marcos Sebastião de Paula Gomes, Hugo Gomes da Silva, Bernardo Caroli de Souza, Sergio R. K. Morikawa

**Affiliations:** 1Mechanical Engineering Department, Pontifical Catholic University of Rio de Janeiro, Rua Marquês de São Vicente 225, 22453-900 Rio de Janeiro, Brazil; mspgomes@puc-rio.br (M.S.d.P.G.); hugomes@puc-rio.br (H.G.d.S.); bcarolid@aluno.puc-rio.br (B.C.d.S.); 2Ouro Negro SA, Rua General Argolo 61, 20921-394 Rio de Janeiro, Brazil; luiz.guedes@ouronegro.com.br; 3Petrobras Research and Development Center (CENPES), 21941-915 Rio de Janeiro, Brazil; morikawa@petrobras.com.br

**Keywords:** gas sensor, photonic crystal fiber, tunable laser, fiber bragg grating, carbon dioxide monitoring

## Abstract

A realistic implementation of an all-fiber CO_2_ sensor, using 74 cm of hollow core photonic crystal fiber (HC-PCF) as the cavity for light/gas interaction, has been implemented. It is based on CO_2_ absorbance in the 2 µm region. The working range is from 2% to 100% CO_2_ concentration at 1 atm total pressure and the response time obtained was 10 min. Depending on the concentration level, the sensor operates at one of three different wavelengths (2003.5 nm, 1997.0 nm and 1954.5 nm) to maintain a high sensitivity across all the working range.

## 1. Introduction

Industrial applications for gas sensing technologies frequently require the sensing system to be immune to interference, to resist harsh environments, to be safe regarding the risk of fires and explosions, and the sensors are often installed in such a way that the communication between them and the recording equipment cannot be processed in a wireless mode. For example, in the oil industry, the measurement of carbon dioxide concentrations in carbon capture and geologic storage operations (CCS) may require the sensor to be installed deep in a CO_2_ injection well, or inside a surface oil-gas separation equipment working in high pressures, creating great difficulties with respect to the transmission of the sensor signal. The same restrictions may apply to other situations of practical interest in several industries, in which fiber optic sensors may be selected as the more appropriate option. Qiao et al. calls attention to the necessity of the oil and gas industry with respect to advances in technology for cost-effective production in new areas [1]. Applications are described for fiber Bragg gratings (FBGs) [2] sensors in the well-logging field, including seismic explorations. The environment in which the sensors have to perform is extremely severe, and the advantages of using fiber optic sensors in such circumstances are enumerated in comparison with other types of technologies.

Fiber optic sensors are less prone to interference, and, depending on the wavelength used, the fiber provides an efficient way to communicate the signal at distances of several kilometers. In principle, it can be compact and light weight and does not require electric power near the measurement location. For gas measurement, several spectroscopic techniques can be adapted using fiber to guide light to the measurement location, such as Raman, absorbance, and interference. Hollow-core photonic crystal fibers (HC-PCF) present attractive promises for the implementation of these techniques as it provides excellent overlap of light and gas inside its hollow core, with a long interaction length and reduced size and weight.

Several authors have demonstrated many configurations using microstructure fibers for gas detection [3]. Cubillas et al. [4] demonstrated an HC-PCF as a gas cell for methane detection at 1.3 µm. An investigation into the feasibility of using microestructured fiber in acetylene gas detection in the 1.5 µm region was reported by Ritari et al. [5]. Yang et al. [6] described a Raman spectroscopy system for the detection of several gases, including carbon dioxide and vapors, using an HC-PCF probe for gas sensing in environmental control applications. Heidari et al. [7] described a miniaturized microstructure spectroscopic gas sensor using a tunable slow-light HC-PCF, and called attention to the enhanced electrical field, localized in the hollow-core of the fiber, due to the tunability of slow-light modes. Other approaches have also been implemented. Exploring photonic crystal fibers, Villatoro et al. [8] demonstrated a photonic crystal fiber (PCF) interferometer system for chemical vapor detection. Nevertheless, HC-PCF has drawbacks that have prevented its use in commercial sensing systems. Among these problems are the difficulty of efficiently coupling light and gas to its core, the long response time, and the presence of unstable guided high order modes [9].

Optical absorption [10], due to molecular transition in gases, is a linear optical property that can be easily measured and is independent of absolute power fluctuations. Gases, in general, present absorption spectra, which depend on its molecular structure and are a signature of each gas. Although the spectra of different gases overlap over some wavelength regions, there are specific wavelengths where absorption can be associated with only one of the most common atmospheric gases. In particular, CO_2_ presents absorption lines at 1954.5 nm, 1997.0 nm, and 2003.5 nm, as can be observed in Figure 1, which do not overlap with water vapor or other important atmospheric gases.

In a fixed length cavity, the absorption, or the non-absorbed transmitted light, at these chosen wavelengths, can be correlated with the concentration of a gas species via the Beer Lambert Law (Equation (1)), which predicts the transmitted light intensity, $I_{x}$, through an absorbing medium by:(1)$$I_{\lambda }\left(x\right)=I_{o}\times \exp^{-a\left(\lambda \right)x},$$
where *x* is the distance traveled by light across a homogenous medium with an attenuation coefficient, a(λ). In gases, a is proportional to the absorbing gas concentration [11].

When selecting the wavelength to be used to sense CO_2_ in a fiber optic system, one has to consider several aspects, such as the interaction length, in this case the HC-PCF length; the total optical fiber length; and the absorption intensity. Figure 1 shows the light transmittance across 74 cm of CO_2_ at 20% concentration and a total pressure of 1 atm. At 1580 nm, telecom fibers present extremely low attenuation and all optical components are readily available, but, in this wavelength region, the weak absorption lines of CO_2_ would demand a very long interaction length. Other options are to explore the absorption lines near 1950 nm and 2000 nm. In this case, the drawback is that SiO_2_ based fibers present attenuation that increases steeply beyond 1800 nm reaching 10 dB/km at 1950 nm and 20 dB/km at 2000 nm [12,13]. The choice between these two wavelength regions will depend on the specific application, including the total length of fiber and the optical power of the interrogator.

In this work, a complete, rugged all-fiber CO_2_ sensing system is presented based on the transmitted light in the 2000 nm, 1954.5 nm, 1997.0 nm, and 2003.5 nm spectral region. The optical cavity, where light and gas interact, is made of 74 cm of HC-PCF. The system measures the optical transmission near specific absorption lines of CO_2_, which are automatically selected depending on the CO_2_ concentration. By using different CO_2_ lines, it is possible to cover a wide concentration range (2% to 100%) at atmospheric pressure. The light source used is a tunable laser (Cr^2+^:ZnS/Se) with an output power on the order of 1 W. Its high power allows it to reach distances of more than 1 km from the measurement point.

## 2. Sensor Design and FBG Tuned Laser

### 2.1. Working Principle

To quantify the CO_2_ concentration, we used the Hitran database to simulate the transmission across the length of the optical cavity of the sensor (74 cm) [14], which means considering *x* in Equation (1) as equal to 74 cm. These results were further processed, as explained ahead, to be directly comparable to the measured quantities. As an example, Figure 2a presents the transmittance, given by Hitran, of around 2003.5 nm for CO_2_ concentrations of 5% and 20%. It is clear that for the increasing concentration, not only is the minimum transmission point reduced, but also the lateral maxima, on both sides, do not reach 100% transmission. In this work, the difference between the transmission of the lateral maxima and the minimum transmission level is called the “transmission valley”. These curves were normalized, dividing it by the local maximum, giving rise to a “normalized transmission valley” as shown in Figure 2b. The minima of these curves were obtained for several concentration levels at three wavelengths (1954.5 nm, 1997.0 nm, and 2003.5 nm) [14]; this simulated data is plotted in Figure 2. Transmittance around 2003.5 nm for CO2 concentrations of 5% and 20% across a 74 cm long cavity at total pressure of 1 atm. It is important to point out that the sensor measures the transmitted light in the vicinity of one absorption line and, consequently, it does not measure 100% transmission in the scanning range. This normalized transmission valley is the quantity which is measured by the sensing system and can be directly compared with the graphs shown in Figure 2c.

The curves of Figure 2c are described by the following Equations (2)–(4), for 1954.5 nm, 1997.0 nm, and 2003.5 nm, respectively:(2)T1954.5=92.85exp(−C31.79)+6.95,
(3)T1997.0=81.97exp(−C90.60)+18.02,
(4)T2003.5=99.01exp(−C10.73)+0.58,
where T is the normalized transmittance for each wavelength and C is the CO_2_ concentration. Thus, the sensitivity is the derivative of these curves. For concentrations below 20%, the 2003.5 nm line presents the highest sensitivity, but it saturates at higher concentrations. In order to have a good sensitivity up to 100% CO_2_ levels, depending on the concentration level, the system must be operated either at 1997.0 nm or at 1954.5 nm. All three lines were chosen based on their strength and the absence of water vapor lines in its spectral vicinity.

### 2.2. Sensor Design

A schematic view of the complete system is presented in Figure 3. It is based on a dual path correlation scheme [15,16], where the laser light is split in two arms, one used as a power reference while the other carries light to the optical cavity where light/gas interaction takes place. Dividing the output of the detector receiving light that has passed through the cavity by the reference detector, all power fluctuations from the laser are canceled. If the laser wavelength is kept constant, the resulting ratio of the two optical signals is basically constant. Once the laser is set to sweep back and forth, the ratio of the two signals will indicate the spectral response of the optical cavity, including the gas under investigation.

Equations (5) and (6) describe the output voltage for $V_{1}$ and $V_{2}$ for the detector receiving light from the gas cavity, and for the reference arm, respectively:(5)$$V_{1}=P\times L_{c1}\times L_{f1}\times G_{1}\times L_{\mathrm{pcf}},$$
(6)$$V_{2}=P\times L_{c2}\times L_{f2}\times G_{2},$$
where P is the laser output power coupled to the fiber, $L_{c1}$ and $L_{c2}$ are the coupling ratio losses at the fiber coupler, and $L_{f1}$ and $L_{f2}$ are the intrinsic fiber losses for each arm. $G_{1}$ and $G_{2}$ are the opto-electronic conversion factors and $L_{\mathrm{pcf}}$ is a combination of the cavity insertion losses and gas absorption. As $L_{c1}$, $L_{c2}$, $L_{f1}$, $L_{f2}$, $G_{2}$, and $G_{2}$ are constants, the ratio, R, between $V_{1}$ and $V_{2}$ can be expressed as Equation (7):(7)$$R=\frac{V_{1}}{V_{2}}=K\times L_{\mathrm{pcf}},$$

With K constant and $L_{\mathrm{pcf}}$ given by the product of the insertion losses, $L_{i}$, and the gas absorption, A, which is intended to be measured as:(8)$$L_{\mathrm{pcf}}=L_{i}\times A,$$

The signal obtained will have no specific unit as it is the ratio of two optical power levels and depends on losses in both arms of the sensors as well as the electronic gains of the two photodetectors. All these factors are constant, or vary very slowly compared to the time of each measurement. To compare the measured optical spectra equivalent to those presented in Figure 2b, the signal must be normalized, which means to set the lateral maximum level, beside the transmission valley, to 100%.

It is important to point out that this measurement is independent of the laser power and the exact laser wavelength is not important; as long as the laser is scanned through the minimum transmission point, the software will pick the correct value. The only source of error is to establish the value that should be used as 100% to normalize the transmission valley.

This system can be used with different lasers, as long as it can be coupled to the optical fiber and can be scanned across the spectral regions of interest.

### 2.3. Implementation

As depicted in Figure 4b, light is launched in the HC-PCF from a single mode fiber (SMF 28) and collected with a multi-mode fiber. In both sides, a gap of a few microns allows the gas to enter the sensing region. As can be seen in Figure 4a, the gas inlet and outlet are placed inside small boxes specially designed to accommodate an alignment base. These boxes are hermetically sealed with a connection for a tube. The inlet tube is open and can be placed where the gas must be collected, while the outlet tube is connected to a remote small pumping system that maintains a negative pressure to force the gas into the HC-PCF. This scheme reduces the response time of the sensor to 10 min, as can be visualized in the results presented in the next section.

It is important to point out that the pumping system is necessary to reach a response time of a few minutes. As demonstrated by Hoo et al. [16], the diffusion time for gases in fibers with similar geometry is in the order of one minute for a 7 cm long fiber and can be extrapolated to more than one hour for a fiber length of 74 cm.

The system shown in Figure 4a was implemented with the following main components:Optical cavity: The optical cavity is composed of a 74 cm long HC-PCF (NKT HC 2000);tunable laser: The laser used to obtain the results presented in the next section was a single-frequency tunable Cr^2+^:ZnS/Se laser from IPG Photonics Corporation, with an output power up to 2 W. Its spectral linewidth is 0.7 MHz, which is equivalent to 0.01 pm at 2 micron and the tuning range is adjustable from 1933 nm to 2245 nm. The scanning speed can be selected between 0.04 nm/s and 35.70 nm/s, with discrete steps of 0.14 pm;fibers and coupler: Light is launched in the HC-PCF from a single mode fiber (SMF 28) and collected with a multi-mode fiber. The fiber coupler is designed to operate as 50/50 at 2 µm; andprocessing unit: The acquisition and processing unit uses a National Instruments NI 6009 DAC board (National Instruments, Austin, TX, USA) [17], connected to a PC running a program written in Labview. The program is responsible for all the digital control and processing, going from the DAC board settings, all the way to the user interface. Figure 5 shows the end user interface where information on the CO_2_ concentration history and optical signal can be visualized.

The transmittance measurements are converted to CO_2_ concentrations and shown in the user interface panel. The system offers an option of continuously averaging a number of measured points, which can be chosen by the user. The latest value, as well as the CO_2_ concentration history, is presented in the main panel. As a default, the system saves all raw data obtained which, if necessary, can later be reprocessed with different settings.

Depending on which wavelength the laser is operating, the processing unit compares the measured transmittance to the appropriate curve to calculate the CO_2_ concentration using Equations (2)–(4). It is important to point out that the working principle of this implementation does not require any calibration, making it very stable and not influenced by variations in the absolute power level.

## 3. Results and Discussion

The system is extremely flexible and, depending on the specific application, some operational parameters can be changed to achieve the best performance. For the results presented in this section, the following parameters were used:Wavelength span: 0.6 nm (2003.1 nm to 2003.7 nm);sweep speed: 0.2 nm/s; andmeasurement update interval: 12 s.

With these settings, every 12 s, the system reads four times the same absorption line and processes it, presenting the average transmittance at maximum attenuation. Individually, each of the four measurements corresponds to a trace similar to those shown in Figure 6.

To characterize the overall sensor’s performance when exposed to different CO_2_ concentrations, it was exposed to cycles of pure N_2_ and 5%, 10%, 15%, and 20% of CO_2_. Figure 7a shows the transmittance at the 2003.5 nm during these gas cycles. As observed, during these cycles, the response time of the system was approximately 10 min, see Figure 7b. As can be seen, in the typical signal shown in Figure 6, non-fundamental modes introduce a signal background fluctuation with a spectral frequency that cannot be separated from the optical signal of interest. The amplitude of these fluctuations was in the order of 10% of the total signal, which translates into, approximately, 1.5% of the CO_2_ concentration when using the 2003.5 nm absorption line. This effect explains why the maximum transmittance level shown in Figure 7a is not 100% for pure N_2_, and determines that the minimum detectable concentration is around 2%.

As mentioned earlier the processing software translates the measured transmittance into CO_2_ concentration using curves shown in Figure 2c. The final performance of the sensor can be appreciated in Figure 8, where its output is compared with the reference gas concentration. Measurements were performed at 2003.5 nm and 1954.5 nm for concentrations of up to 20% CO_2_. The 1997.0 nm line was used for CO_2_ concentrations of 20% and 100%.

The presence of higher order modes not only limits the minimum detectable CO_2_ concentration, but also introduces measurement uncertainties. As the attenuation of light affects all transmission modes, for higher concentration levels, the modal interference pattern was reduced and a better a resolution was obtained. Using an average of 10 points for each final measure, the mean error was 0.4% of CO_2_ concentration with a maximum error of 0.7%.

The intermodal interference presents an amplitude that is independent of the HC-PCF length. This leads to a tradeoff between the sensitivity and response time associated with the HC-PCF cavity length. The response time increases with the length of the cavity [18] while the sensitivity is inversely proportional to it. If the cavity length were increased by a factor of 10, to 7.4 m, the minimum detectable CO_2_ concentration would be 0.2%, but the estimated response time would be more than one hour. On the other hand, if the HC-PCF length were reduced to 7.4 cm, the response time would be just a few seconds, but the minimum detectable concentration would be 20% of CO_2_. Considering the typical industrial applications in which this system is expected to be used, the most adequate cavity length is in the order of one meter.

## 4. Conclusions

Due to its all fiber configuration, the sensing system presented was mechanically very stable, and its working principal makes it independent of recalibrations, thus it is suitable for long period applications. Even though the fiber attenuation was in the range of 10 dB/km to 20 dB/km, the high power of the light source used enables applications at distances beyond 1 km between the power source and the measurement point.

Although the accuracy of the sensor was better than 0.5% of CO_2_ concentration, the presence of high order modes limited the minimum detectable CO_2_ level to 2%. Without any significant change, but at the expense of increasing the response time, longer HC-PCF could be employed, reducing the minimum detectable level. If improved fibers, with lower mode interference, were available, the system would operate from much lower CO_2_ levels all the way to pure CO_2_.

## Figures and Tables

**Figure 1 sensors-18-04393-f001:**
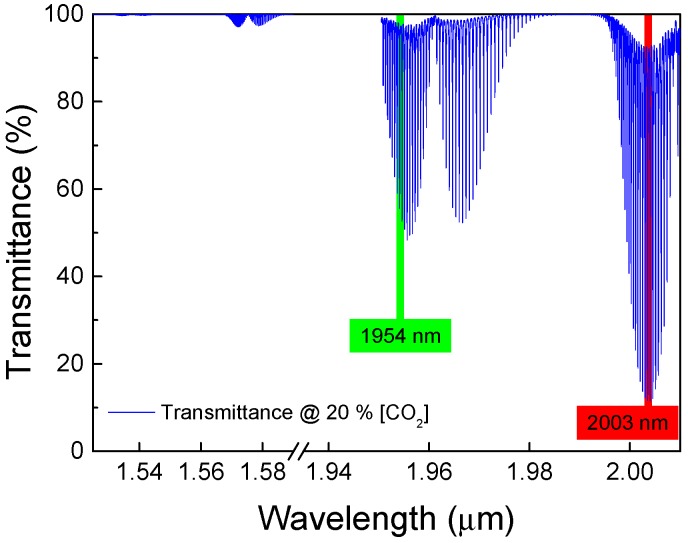
Transmittance for 20% CO_2_ concentration and total pressure of 1 atm in a 74 cm optical path.

**Figure 2 sensors-18-04393-f002:**
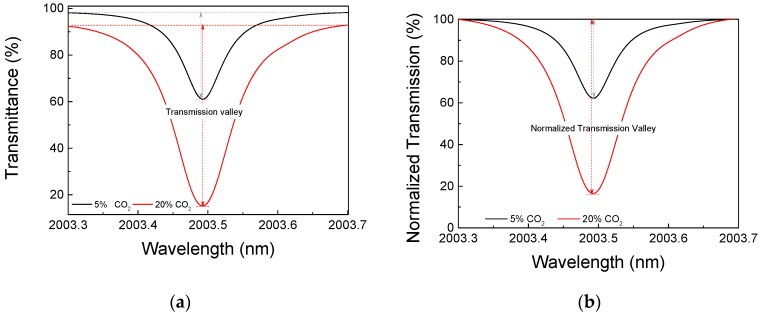
(**a**) Transmittance around 2003.5 nm for CO_2_ concentrations of 5% and 20% across a 74 cm long cavity at total pressure of 1 atm; (**b**) normalized transmission; (**c**) normalized transmission valley as a function of CO_2_ concentration.

**Figure 3 sensors-18-04393-f003:**
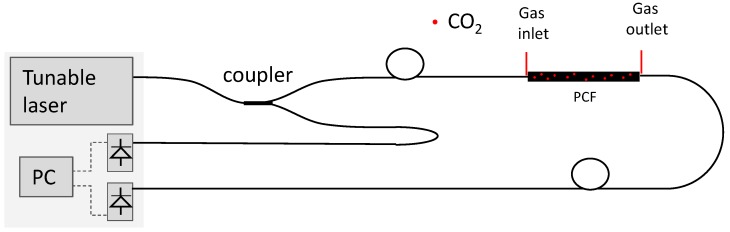
Complete sensor schematic setup, including the HC-PCF based gas/light interaction cavity and portable computer (PC) based acquisition and processing unit.

**Figure 4 sensors-18-04393-f004:**
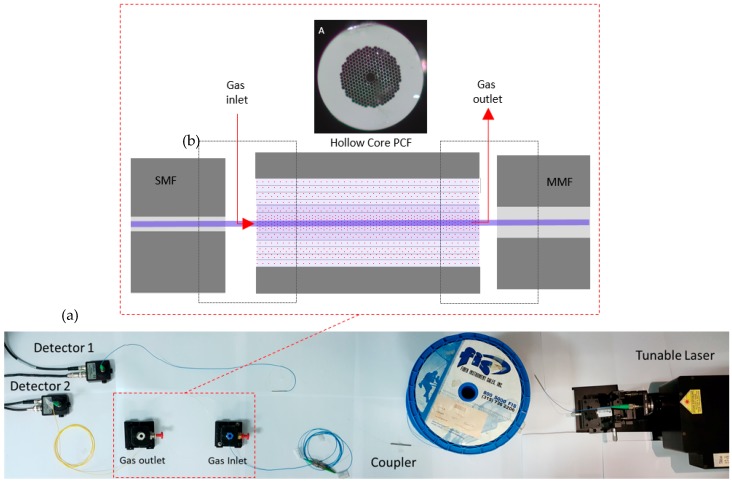
(**a**) Photograph of the complete sensor setup; (**b**) schematic representation of the injection and extraction points for gas and light from the sensing HC-PCF. In detail, the cross section of the HC-PCF used.

**Figure 5 sensors-18-04393-f005:**
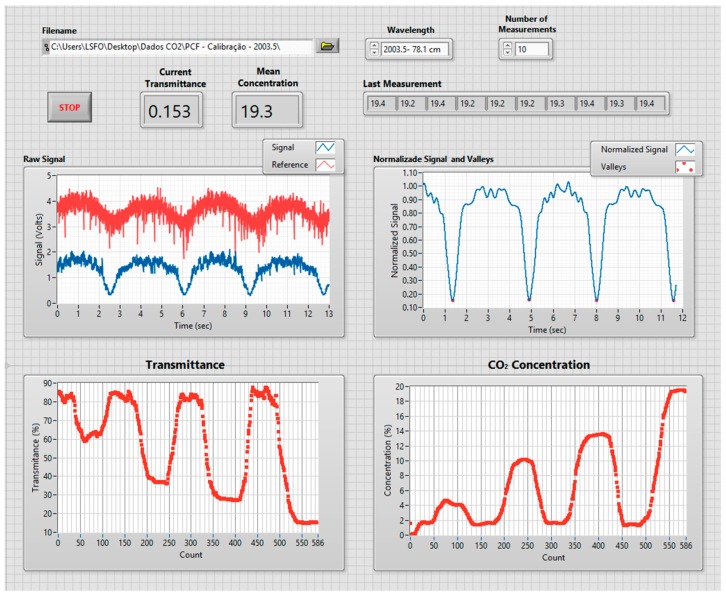
Computer interface of the acquisition and processing unit of the sensor.

**Figure 6 sensors-18-04393-f006:**
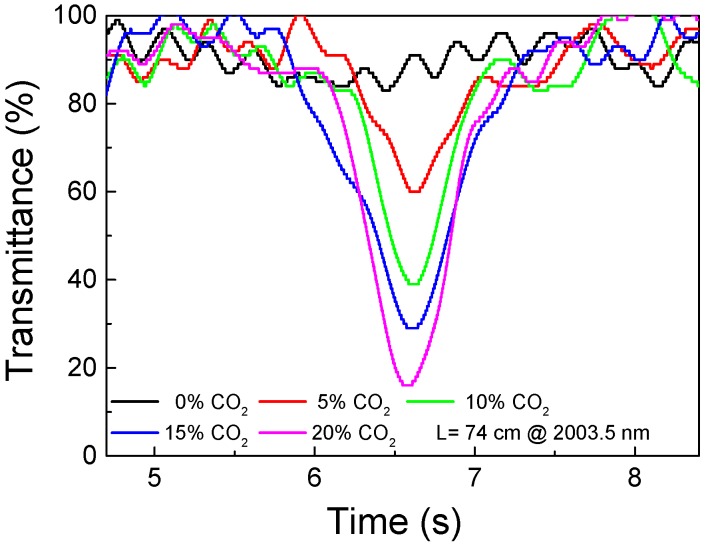
Typical normalized optical signal at different concentration levels.

**Figure 7 sensors-18-04393-f007:**
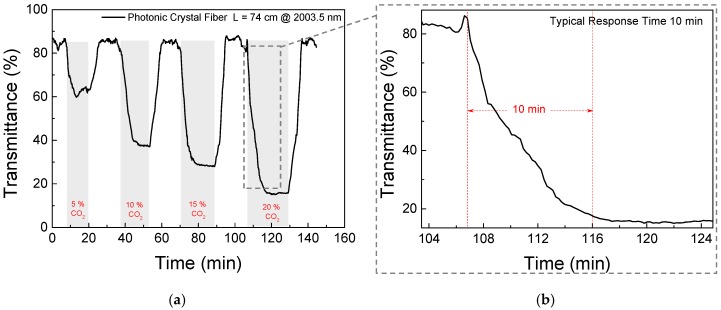
(**a**) Transmittance measured for cycles of different CO_2_ concentration (0%, 5%, 10%, 15%, and 20%); (**b**) the response time of the system.

**Figure 8 sensors-18-04393-f008:**
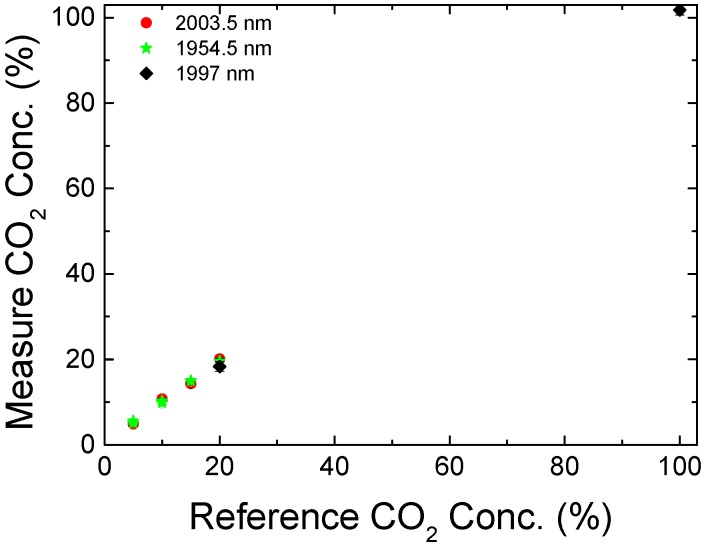
Final CO_2_ concentration indicated by the sensor compared with nominal concentrations of the measured gas.

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
