# Peer review of "All-Fiber CO2 Sensor Using Hollow Core PCF Operating in the 2 µm Region"

_sensors, 2018, doi:10.3390/s18124393_

Reviewer 1 Report

Comments to the Editor:

Authors present a CO2 sensor based on a tunable laser and a Hollow Core fiber (PCF). I consider that the work have some interesting aspects such as the gas cell design. However authors failed to explain with detail their main achievements. Here I consider that the paper was poorly structured and authors must provide more information supporting the principle of operation of their sensor. Therefore I suggest to consider the paper for publication after major revisions that include a restructure of the paper, enhancement of the literature revision presented in the introduction and a strong technical support of the principle of operation of the sensor.

Authors present a CO2 sensor based on a tunable laser and a Hollow Core fiber (PCF). I consider that the work have some interesting aspects such as the gas cell design. However authors failed to explain with detail their main achievements. Here I consider that the paper was poorly structured and authors must provide more information supporting the principle of operation of their sensor. Therefore I suggest to consider the paper for publication after major revisions that include a restructure of the paper, enhancement of the literature revision presented in the introduction and a strong technical support of the principle of operation of the sensor.

Some specific aspects that I consider should be addressed by authors are the following.
1. In the introduction it is necessary to clarify what is the sensing technique that you will use, and provide some references related with the topic. Afterwards, describe advantages and disadvantages of this technique and based on the references provide some qualitative examples of the technical characteristics that have been achieved by similar sensors.
2. In the last paragraph of the introduction please clarify and focus only in the description of the most important aspects of your sensor.
3. Now about the sensor configuration setup, which is based on a laser/beam splitter/two paths/two detectors, it is not a new design and it has been used since long time ago, therefore I suggest to include some references, such as:
a)  Wei, W., et al. (2017). "Water vapor concentration measurements using TDALS with wavelength modulation spectroscopy at varying pressures." Sensor Review 37(2): 172-179.
b)  Vargas-Rodriguez, E., et al. (2017). "Gas Sensor Design Based on a Line Locked Tunable Fiber Laser and the Dual Path Correlation Spectroscopy Method." Applied Sciences 7(9): 958.
c)  Hoo, Y. L., et al. (2010). "Fast Response Microstructured Optical Fiber Methane Sensor With Multiple Side-Openings." IEEE Photonics Technology Letters 22(5): 296-298.
4. In section 2 I would expect to get a specific subsection where the principle of operation of the sensor is described in detail. For instance, what is the purpose of the 2 detectors?, why two channels are formed (reference and measurement)?. Please, include an equation that mathematically describe the relationship between the 2 detectors.
5. In your manuscript you consider a 74 cm fiber, which is the pathlength of the gas cell. The question is why you select this length?. Here, if we consider that you are trying to detect high concentrations levels (≈2-100%) it can be better use a shorter length otherwise as you mentioned in your manuscript the sensor rapidly will “became” saturated. So, why consider this length?, please explain this.
6. In each figure that contain more than 1 graph please labeled (a, b, c, …) and include their corresponding  sub-captions. Afterwards in the manuscript “cite” them as for example Figure 6a.
7. In some parts of the manuscripts appears CO2 please change to CO2
8. It is necessary to describe the full sensor setup and particularly the laser setup implemented for the application. The setup presented in Figure 6 it is incomplete, for instance it is not show where the Single-Frequency Tunable Cr2+:ZnS/Se is placed and neither presented where the Thulium fiber laser is, nor how these are related (if these are). Please explain with detail how the tunable laser is working.
9. In Figure 4, I suggest to simulate the transmittance for the line at 2003.5 nm up to 100%, as for the other 2 cases presented.
10. In line 178 authors stated that the transmittance did not reach to 100% when the CO2 concentration is 0% since there is background fluctuations, however this is not clear for me if these fluctuations are due to the background why these are not strongly minimized by the 2 detectors, since generally this is the main advantaging of use them. So please clarify this issue.
11. About the characteristics of the tunable laser, please state which are the wavelength resolution or wavelength steps achieved by the laser?
12. In figure 8 authors presented the transmittance, as far as I can understand the considered just the peak occurring at certain wavelength (2003.5), please clarify if it is true?. This is quite important since can affect the wavelength accuracy, since any error in the wavelength position can induce to large errors in the concentration measurements. Here, why not consider the integral under the curve, the RMS to minimize errors due to wavelength accuracy. Here I suggest to clarify this issue and include a mathematical equation that describe how the concentration is being calculated.
13. The sensor response time from my point of view is quite large considering that you are measuring lager CO2 concentrations, please compare this figure with similar the obtained with similar sensors, and explain why your is large, how it is related with the length of the PCF. Moreover, consider if the length of the fiber is reduced considerably can this reduce the response time of the sensor, without affecting the dynamic CO2 concentration range (2-100%) and having just one PCF length.
14. In some parts of the manuscripts where authors stated numbers they used periods (.) instead of (,). This need to be corrected accordingly.
15. In general the English edition must be carefully checked and corrected where necessary.

Author Response

Dear Referee,

I'm sending to you the revised version of our manuscript entitled “All-fiber CO2 sensor using Hollow Core PCF operating in the 2 µm region", for your appreciation.

We carried out a complete review of the manuscript following your suggestions and answering to all your questions. All the changes are highlighted in the revised manuscript. Furthermore, in the attached document we answered in detail all points that you had mentioned, improving the manuscript where it was necessary.

We hope this improved version could be accepted to be published in this Journal.

Reviewer 2 Report

The paper is very good. The experiment is precise and the results are reliable. I had some small questions but in the course of my reading the authors give very good answers

In this manuscript, the authors explain about “All-fiber CO2 sensor using Hollow Core PCF operating in the 2 mm region”. Manuscript is well written and findings are interesting which helps to for CO2 sensor, however it requires some refinement to improve quality of the manuscript.

Main comments

1.  Abstract:  What is FBG?  Do not use any abbreviations without any explanations in advance.
2.  Please check the font size throughout the manuscript, it should be same.
3.  There are some grammatical and syntax error, please check and correct throughout the manuscript
4.  Please include real setup of gas cell structure of hollow-core photonic crystal fiber to detect trace gases
5.   In the experimental section, did the author obtain the theoretical absorption spectra using Spectral Calculator?
6.    What about selectivity? Did the author try other gases to check selective nature?

Author Response

Dear Referee,

I'm sending to you the revised version of our manuscript entitled “All-fiber CO2 sensor using Hollow Core PCF operating in the 2 µm region", for your appreciation.

We carried out a complete review of the manuscript following your suggestions and answering to all your questions. All the changes are highlighted in the revised manuscript. Furthermore, in the attached document we answered in detail all points that you had mentioned, improving the manuscript where it was necessary.

We hope this improved version could be accepted to be published in this Journal.

Round  2

Reviewer 1 Report

Authors have improved considerably the manuscript and have attended many of the recommendations of the reviewers. However, in the new version there are some point that have been introduced that requires further improvements and clarifications. Hence, I suggest to consider the paper for publication after major revisions of the paper are performed. Some specific aspects that I consider should be addressed by authors are the following.

1.  For clarity purposes I suggest to rewrite equation 1, stating the variables that you will use later in the manuscripts. For instance if you will use L for define the gas pathlength using this letter in the equation. Moreover the absorption is defined as the gas concentration times the monochromatic absorption coefficient (which is taken form HITRAN) and it is wavelength depend. So I suggest to rewrite this equation using this parameters, since later you use. Recall that the light absorbed by the gas in the cell is practically given as A=1-I0/Ix.
2.  Rewrite 107 and 108, since right now it is confusing.
3.  In equations 2-4 please write in normal style (not italics) the exp, since it is a function.
4.  In 5 and 6, I suggest to eliminate the * operator symbols and if it is completely necessary please use the operator symbol times (×), in order to avoid confusions. Moreover, in your variables use subscripts Lc1, Lpcf, etc. Also include a comma after each equation.
5.  It is not clear form me how you jumped from 7 to 8, please provide more details. Also it is necessary to define what means A, C and L (please check my first comment).
6.  In line 198 there is an error with the figure label.
7.  In line 187 and elsewhere in the manuscript: to point that à to point out that
8.  In line 206: discreet à discrete
9.  I would recommend to eliminate the section 2.4 since you are describing a tunable laser that still under study. So for clarity purposes I strongly recommend to eliminate this section of the manuscript, you can consider this information as a part of a future work.
10. In the results section I would like to have more information about how do you calculate the sensitivity and the minimum detectable concentration. Also what about the resolution of the sensor? It is the same in all the full dynamic range of concentration (0<=C<=100%). It is important to state since the sensor used 3 different lines to measure different concentration ranges.  Therefore it is necessary to provide more details and discuss these issues.

I consider that it is necessary to provide a further explanation of why do you normalized the “transmission valleys”. I understand that the lateral maximum as the concentration increases the minimum transmission (deep valley) will go to cero and will keep at this level , however the lateral maximum will still decreasing. However, I consider that is missing a brief and clear explanation of your motivation to perform this normalization and what are the advantages. Please, provide more details about what are the advantages of carry out this operation.

Author Response

Dear Referee,

I'm sending to you the second revised version of our manuscript entitled “All-fiber CO2 sensor using Hollow Core PCF operating in the 2 µm region", for your appreciation.

We carried out a complete review of the manuscript following your suggestions and answering to all your questions. All the changes are highlighted (green) in the revised manuscript. Furthermore, we answered in detail all your points mentioned, improving the manuscript where it was necessary.
